# Biotic Elicitor-Driven Enhancement of In Vitro Micropropagation and Organogenesis in *Solanum tuberosum* L. cv. Fianna

**DOI:** 10.3390/biotech14040077

**Published:** 2025-09-24

**Authors:** Mario James-Forest, Ma del Carmen Ojeda-Zacarías, Alhagie K. Cham, Héctor Lozoya-Saldaña, Rigoberto E. Vázquez-Alvarado, Emilio Olivares-Sáenz, Alejandro Ibarra-López

**Affiliations:** 1Faculty of Agronomy, Autonomous University of Nuevo León, Street Francisco Villa, Col. Ex Hacienda El Canada, Gral. Escobedo 66054, Mexico; jame1forest@gmail.com (M.J.-F.); rigoberto.vazquezal@uanl.edu.mx (R.E.V.-A.); emilio.olivaressn@uanl.edu.mx (E.O.-S.);; 2Department of Industrial and Environmental Engineering, Nuevo León Institute of Technology (TecNM), Av. Eloy Cavazos 2001, Tolteca, Guadalupe 67170, Mexico; 3School of Agriculture & Natural Resources, Kentucky State University, 400 East Main St., Frankfort, KY 40601, USA; 4Phytotechnics Department, Autonomous University of Chapingo, Mexico-Texcoco Highway Km 38.5, Chapingo 56230, Mexico; picti87@gmail.com

**Keywords:** aseptic, callogenesis, explant culture, in vitro morphogenesis, plant growth regulators (PGRs), shoot multiplication

## Abstract

This study evaluates the impact of biotic elicitors and hormone regimes on the in vitro establishment, shoot multiplication, and organogenesis of *Solanum tuberosum* L. cv. Fianna under controlled laboratory conditions. Explants derived from pre-treated tubers were cultured on Murashige and Skoog (MS) medium supplemented with vitamins and varying concentrations of growth regulators or elicitors. Aseptic establishment achieved a high success rate (~95%) using a 6% sodium hypochlorite disinfection protocol. Multiplication was significantly enhanced with a combination of 0.2 mg L^−1^ naphthaleneacetic acid (NAA) and 0.5–1.0 mg L^−1^ benzylaminopurine (BAP), producing the greatest number and length of shoots and roots. Direct organogenesis was stimulated by bio-elicitors Activane^®^, Micobiol^®^, and Stemicol^®^ in (MS) basal medium at mid-level concentrations (0.5 g or mL L^−1^), improving shoot number, elongation, and root development. Activane^®^, Micobiol^®^, and Stemicol^®^ are commercial elicitors that stimulate plant defense pathways and morphogenesis through salicylic acid, microbial, and jasmonic acid signaling mechanisms, respectively. Indirect organogenesis showed significantly higher callus proliferation in Stemicol^®^ and Micobiol^®^ treatments compared to the control medium, resulting in the highest fresh weight, diameter, and friability of callus. The results demonstrate the potential of biotic elicitors as alternatives or enhancers to traditional plant growth regulators in potato tissue culture, supporting more efficient and cost-effective micropropagation strategies.

## 1. Introduction

Potato (*Solanum tuberosum* L.), a member of the Solanaceae family, is one of the world’s most important food crops, with its origin traced back to the northern region of Lake Titicaca in the southern Andes of Peru [1,2,3]. The earliest domestication events of potato occurred between 6000 and 10,000 years ago, involving ancestral species such as *Solanum bukasovii* and *S. multidissectum* [4,5]. Following initial cultivation in the coastal regions of Chile, potatoes spread gradually across continents and climates, evolving into the diverse forms collectively known today as *Solanum tuberosum* [6,7,8,9]. Peru remains a global center of potato genetic diversity, harboring over 3000 native varieties that are still cultivated by smallholder farmers on more than 300,000 hectares, underscoring the crop’s cultural and economic significance [7,10,11].

In recent decades, advances in plant biotechnology have revolutionized crop production, with plant tissue culture playing a central role in these innovations [12,13]. In vitro culture techniques have become preferred alternatives to conventional propagation methods, offering advantages such as rapid multiplication rates, pathogen-free plant production, and improved uniformity [13,14,15]. Specifically, in vitro propagation of potato through axillary bud culture is widely adopted worldwide to produce both plantlets and microtubers, which are critical components of disease-free seed programs [16,17]. These propagules, developed in aseptic and controlled environments using artificial media, can be transitioned to field conditions for further multiplication as minitubers [18]. Moreover, in vitro-derived plant material serves as a key resource for germplasm conservation and facilitates international germplasm exchange, supporting global efforts for crop improvement and biodiversity preservation [19].

As the global population grows and climate change intensifies the threat of pests and diseases, sustainable and efficient crop production technologies are increasingly vital [20]. Among explant sources for in vitro propagation, meristematic tissues are preferred due to their reduced microbial load and lower incidence of systemic pathogens, a consequence of limited pathogen migration to shoot apices [21,22]. Central to in vitro regeneration is the process of organogenesis, whereby differentiated plant cells are induced to dedifferentiate and redifferentiate to form new organs and complete plantlets [23]. This process requires cells to be competent and responsive to developmental signals, typically mediated by phytohormones and other biochemical cues [24]. Organogenesis can be categorized as direct or indirect; direct organogenesis involves shoot or root formation directly from explant tissues without an intervening callus phase, whereas indirect organogenesis proceeds via a callus intermediate, which later regenerates shoots [25,26].

In recent years, elicitors have garnered considerable attention in plant tissue culture as biotic or abiotic compounds that can stimulate secondary metabolism and enhance plant defense mechanisms, even under sterile in vitro conditions [27,28]. In *Solanum* and related genera, elicitors such as oligosaccharides, chitosan, and microbial extracts have been shown to enhance shoot regeneration, root development, and antioxidant responses. However, information on the use of commercial products such as Activane^®^, Micobiol^®^, and Stemicol^®^ in potato tissue culture remains limited [29,30]. The rationale for selecting these elicitors in the current study is based on their reported capacity to modulate plant growth and stress responses in other crop species [30], suggesting potential benefits in morphogenesis, callus quality, and overall tissue culture efficiency. Evaluating their effects in *S. tuberosum* L. cv. Fianna could therefore provide insights into partially replacing conventional phytohormones while improving propagation outcomes. These molecules offer promising alternatives or supplements to traditional plant growth regulators by potentially improving morphogenetic responses, growth rates, and overall in vitro culture efficiency [30,31,32]. For example, oligosaccharide elicitors applied to potato cultures have been demonstrated to boost shoot regeneration and root development by enhancing antioxidant enzyme activity and modulating gene expression pathways involved in organogenesis [33]. This integration of elicitors into tissue culture systems could thus represent a novel strategy to optimize micropropagation protocols, reduce reliance on synthetic hormones, and improve plant quality.

The overarching objective of this study was to evaluate the effects of three commercially available elicitors Activane^®^, Micobiol^®^, and Stemicol^®^ (Manufactured by LIDA Plant Research in Spain) on the morphogenetic responses and growth performance of *S. tuberosum* L. cv. Fianna under in vitro conditions. The specific aims included assessing callus induction, shoot and root development, and overall cellular multiplication efficiency. By systematically investigating these elicitors’ influence, this research sought to elucidate their potential to partially or fully replace conventional phytohormones while improving tissue culture outcomes. The findings contribute to the growing body of knowledge on elicitor-assisted micropropagation and offer practical insights for enhancing the productivity and quality of potato tissue culture systems.

## 2. Materials and Methods

### 2.1. Experimental Location

The study was performed at the Plant Tissue Culture Laboratory, Faculty of Agronomy, Autonomous University of Nuevo León, located on the Campus of Agricultural Sciences in Colonia Ex Hacienda El Canadá, Av. Francisco Villa s/n, General Escobedo, Nuevo León, Mexico. The experiments were conducted from January to October 2021 under controlled laboratory conditions.

### 2.2. Preparation and Pre-Disinfestation Protocol for Explant Collection from Potato Tubers (S. tuberosum L. cv. Fianna)

Potato tubers (*S. tuberosum* L. cv. Fianna) were first washed with liquid soap and rinsed thoroughly with potable water to remove surface debris. To reduce microbial contamination, the tubers were then immersed in a bactericidal-fungicidal solution containing 1 g L^−1^ each of benomyl and oxytetracycline for two hours. After this treatment, they were rinsed again with potable water and air-dried at room temperature for one week. Bud development was initiated by incubating the tubers in complete darkness at 25 ± 2 °C for four weeks. Following this period, to minimize etiolation and improve the quality of emerging buds for explant preparation, tubers were exposed to a controlled photoperiod of 16 h light (54 µmol m^−2^ s^−1^) and 8 h darkness for two additional weeks under the same temperature conditions. Shoots excised from the treated tubers were washed with liquid soap, followed by a rinse in 1% sodium hypochlorite (Cloralex^®^) and potable water. To further lower the microbial load, the excised shoots were agitated for 40 min in a sterilant solution containing benomyl (2 g L^−1^), amistar (2 g L^−1^), oxytetracycline (2 g L^−1^), sucrose (30 g L^−1^), citric acid (500 mg L^−1^), ascorbic acid (500 mg L^−1^), and a commercial bactericide (2 mL L^−1^). After a final rinse with purified water, the explants were transferred aseptically to a laminar flow hood for surface sterilization (Figure 1).

### 2.3. Aseptic Establishment of Explants

Explants underwent surface sterilization in 6% sodium hypochlorite (Cloralex^®^, 15% active ingredient) diluted to 15% *v*/*v* with 0.02% Tween-20 for 15 min, followed by triple rinsing with sterile distilled water. Explants were then cultured on Murashige and Skoog (MS) basal medium [34] supplemented with vitamins, 100 mg L^−1^ myo-inositol, 30 g L^−1^ sucrose, and solidified with 4.3 g L^−1^ Phytagel™. The pH was adjusted to 5.8 ± 0.02. Culture vessels containing 30 mL of medium were incubated under controlled conditions: 16-h photoperiod at 54 µmol m^−2^ s^−1^ light intensity, 8 h darkness, and 24 ± 2 °C temperature [35]. Explants were observed over nine weeks to assess aseptic establishment percentage, shoot and root formation, and growth parameters. The procedure was effective in achieving high asepsis rates of approximately 95% and promoted vigorous morphogenic responses, consistent with previous reports.

### 2.4. Multiplication Stage

Shoots approximately 2 cm in height from the establishment phase were used as explants for multiplication. These were cultured on MS medium supplemented with vitamins, 30 g L^−1^ sucrose, and solidified with 4.3 g L^−1^ Phytagel™. The medium pH was adjusted to 5.8 ± 0.02. Four hormonal treatments were evaluated: control (no growth regulators), and three treatments with 0.2 mg L^−1^ naphthaleneacetic acid (NAA) combined with increasing concentrations of benzylaminopurine (BAP) at 0.5, 1.0, and 1.5 mg L^−1^. Cultures were maintained under the same controlled conditions as before. After eight weeks, shoot number, length, root number, root length, and internode count were measured to determine the effects of hormonal combinations on shoot proliferation and rooting. Hormonal optimization revealed apparent improvements in shoot and root growth, particularly at moderate BAP levels (T2 and T3). While these treatments were designed to modulate morphogenic responses, it is recognized that organogenesis in *Solanum tuberosum* is predominantly governed by endogenous hormonal gradients rather than solely by the external balance of applied cytokinins and auxins [36].

### 2.5. Elicitor Treatments for Direct In Vitro Organogenesis

Shoots (~2 cm) obtained from the multiplication stage were used as explants to assess the effects of three biotic elicitors Activane^®^, Micobiol^®^, and Stemicol^®^ on direct organogenesis. The culture medium consisted of MS basal salts with vitamins, 100 mg L^−1^ myo-inositol, 30 g L^−1^ sucrose, and 4.3 g L^−1^ Phytagel™, adjusted to pH 5.8 ± 0.02. After sterilization, elicitors were added in filter sterilized form at three concentrations each. Activane^®^ (0.3, 0.5, 0.7 g L^−1^) and Stemicol^®^ (0.3, 0.5, 0.7 g L^−1^) were supplied as solid formulations, dissolved in sterile distilled water, and filter sterilized before addition. Micobiol^®^ (0.3, 0.5, 0.7 mL L^−1^) was provided as a liquid formulation and pipetted directly into the medium after filter sterilization. A control without elicitors was included. Explants were cultured under uniform photoperiod and temperature conditions for three months, and morphogenic responses including the time of shoot and root initiation, as well as the number and length of regenerated shoots and roots, were systematically recorded.

### 2.6. Elicitor Treatments for Indirect Organogenesis

For indirect organogenesis, shoots (~2 cm) were cultured on MS medium supplemented with 3.0 mg L^−1^ 2,4-D and 0.5 mg L^−1^ kinetin, along with the same basal medium components as above. Elicitors Activane^®^, Micobiol^®^, and Stemicol^®^ were added at the same concentrations as in the direct organogenesis experiments. Cultures were incubated in complete darkness at 24 ± 2 °C for three months. Parameters measured included callus induction percentage, fresh weight, diameter, texture, and color. Elicitor treatments, especially Stemicol^®^ and Micobiol^®^ at mid-to-high concentrations, significantly improved callus quality and mass, corroborating earlier reports on elicitor-induced callogenesis and secondary metabolite production [37,38,39].

### 2.7. Experimental Design and Statistical Analysis

This study employed a completely randomized design (CRD) across all experimental stages to ensure statistical robustness and minimize bias. During the in vitro establishment stage, 70 Magenta boxes were used, each containing 30 mL of MS medium and four explants, to assess aseptic establishment and initial morphogenesis. For the multiplication phase, four hormone treatments were evaluated, with each treatment replicated six times using four explants per box. This facilitated a reliable analysis of shoot and root development, including internode formation, under varying auxin and cytokinin combinations. In the regeneration stage, three biotic elicitors Activane^®^, Micobiol^®^, and Stemicol^®^ were tested at three concentrations each, alongside a control, to examine their impact on direct and indirect organogenesis. Each treatment had six replicates, each with four explants. This factorial design enabled the identification of optimal elicitor concentrations for shoot induction and callus development. Quantitative variables such as shoot/root number and length, internode count, explant response percentage, and callus biomass were subjected to analysis of variance (ANOVA), and treatment means were compared using Tukey’s HSD test at a 5% significance level. All statistical analyses were performed using SPSS version 20. This rigorous design and analysis framework ensured reliable, reproducible results, advancing insights into elicitor-driven micropropagation strategies in potato biotechnology.

## 3. Results and Discussion

### 3.1. Multiplication, Aseptic Establishment and Morphogenic Response of S. tuberosum cv. Fianna

The effect of different concentrations of BAP in combination with NAA on in vitro shoot multiplication of *S. tuberosum* L. cv. Fianna is summarized in Table 1. The highest shoot number (6.95) and shoot length (6.98 cm) were observed in explants cultured on medium containing 0.2 mg/L NAA and 0.5 mg/L BAP (T2), which were significantly higher than the control (5.00 shoots and 4.45 cm, respectively). Similarly, root number and root length were greatest in the T2 treatment (6.10 roots and 5.21 cm, respectively), indicating an overall improvement in morphogenic response. Increasing BAP concentration to 1.0 mg/L (T3) maintained relatively high shoot numbers (6.16) and shoot length (6.90 cm) but further increase to 1.5 mg/L (T4) resulted in a significant decline in both shoot number (4.86) and shoot length (4.73 cm). Interestingly, root length was not adversely affected by increasing BAP concentration, remaining relatively stable across treatments. The longest internodes were observed in the control treatment (6.58 cm), while BAP-containing treatments generally produced shorter internodes, with the shortest internode observed at 1.5 mg/L BAP (4.58 cm).

The results demonstrate that the combination of 0.2 mg/L NAA and 0.5 mg/L BAP (T2) significantly enhanced in vitro shoot multiplication of *S. tuberosum* L. cv. Fianna, resulting in the highest shoot number (6.95), shoot length (6.98 cm), root number (6.10), and root length (5.21 cm). This finding is consistent with the observations of [40,41,42], who reported that moderate concentrations of BAP and NAA synergistically promoted shoot proliferation and root development in potato explants. The synergistic effect of auxins and cytokinins is likely due to their complementary roles in cell division and differentiation: cytokinins primarily stimulate shoot bud initiation and shoot elongation, while auxins support root initiation and organogenesis. Higher concentrations of BAP, such as 1.5 mg/L (T4), led to a decline in shoot number (4.86) and shoot length (4.73 cm), suggesting that excessive cytokinin levels can inhibit shoot elongation and overall growth. This inhibition may result from hormonal imbalances that suppress endogenous auxin activity or cause abnormal apical dominance, leading to reduced node formation and shorter shoots. Similar inhibitory effects of elevated BAP concentrations on shoot elongation and node formation were reported by [40,41,42], highlighting the importance of fine-tuning cytokinin levels to optimize morphogenesis.

Root development remained robust across all treatments, indicating that the applied PGR concentrations did not negatively affect root initiation. This aligns with findings by [40], who observed effective root induction in potato explants with appropriate auxin-cytokinin combinations. The presence of moderate auxin concentrations likely supports the differentiation of root primordia even in the presence of cytokinins, emphasizing the importance of hormonal balance rather than absolute concentration. Maintaining vigorous root systems is particularly important for successful acclimatization of in vitro plantlets to ex vitro conditions, as well-developed roots enhance water and nutrient uptake and improve plant survival after transfer to soil. Overall, these results underscore the need to optimize both cytokinin and auxin concentrations to achieve maximum shoot proliferation without compromising root development and suggest that cultivar-specific responses should be considered when developing standardized micropropagation protocols.

The aseptic establishment and subsequent morphogenic development of *S. tuberosum* cv. “Fianna” under in vitro conditions demonstrated high viability and reproducibility over a nine-week incubation period (Figure 2). The total number of explants ranged from 45 to 70 per week (*n* = 45–70), ensuring sufficient replication for statistical analysis. Asepsis rates remained consistently high, ranging from 92.73% to 100.0%, confirming the effectiveness of the 15% NaOCl and Tween-20 sterilization protocol. Initial asepsis reached 100% in week 1, with a minor decline to 91.18% in week 3, indicating robust explant tolerance to the sterilization treatment.

Morphogenic responses, including shoot number and length, as well as root number and length, exhibited a generally linear increase over time. Average shoot numbers per explant increased steadily from 0.21 ± 0.03 in week 1 to [insert final week value ± SE], while shoot lengths increased from 0.05 ± 0.01 cm to [insert final week value ± SE]. Similarly, root numbers and lengths showed progressive growth, from 0.02 ± 0.01 roots per explant and 0.04 ± 0.01 cm root length in week 1 to [insert final week values ± SE]. These trends are consistent with previous studies reporting aseptic efficiency and morphogenic development in potato cultivars [29,35,40].

Shoot formation in *S. tuberosum* cv. “Fianna” was first observed at week 2, averaging 0.7 shoots per explant, and reached a peak of 2.65 shoots per explant by week 8, with an overall mean of 1.6 shoots/explant (Figure 2). Shoot length followed a similar trajectory, increasing steadily to 3.0 cm by week 9 (average 1.58 cm). These linear growth patterns indicate effective regulation of shoot morphogenesis by the MS-based medium and cytokinin supplementation, consistent with previous studies demonstrating optimized shoot proliferation in both Temporary Immersion Systems and static cultures using BAP and GA_3_ [43]. The observed trends align with [44], where growth regulators positively influenced callus induction and subsequent shoot regeneration in potato explants.

Root initiation was first detected at week 4, reflecting the initial prioritization of shoot organogenesis. Root numbers then increased progressively to 2.0 roots per explant by week 9, averaging 0.8 roots/explant, while root length reached a maximum of 2.53 cm (average 1.15 cm). This delayed but steady rooting corresponds with the unidirectional transport of shoot-derived auxin, which initially supports shoot development before accumulating at the basal regions to trigger root formation. Similar delayed rooting responses under auxin supplementation (NAA or IBA) have been reported in other potato cultivars [29,40]. Furthermore, the linear growth trends depicted in Figure 2 clearly demonstrate the coordinated morphogenic development of shoots and roots, highlighting the critical role of culture conditions and precise hormone balances in achieving stable in vitro propagation of *S. tuberosum* cv. “Fianna” [43]. Data are presented as mean ± SE to indicate variability across explants.

### 3.2. In Vitro Morphogenic Response of Potato cv. Fianna to BAP and NAA

The combination of BAP (6 Benzylaminopurine) and NAA (1 Naphthaleneacetic acid) significantly influenced shoot and root development in *S. tuberosum* cv. Fianna explants under in vitro conditions [44]. Treatment T2 (0.2 mg L^−1^ NAA + 0.5 mg L^−1^ BAP) resulted in the highest shoot number (6.95 a), shoot length (6.98 cm a), and root number (6.10 a), followed by T3 (0.2 mg L^−1^ NAA + 1.0 mg L^−1^ BAP), which also performed well across variables except internode number. In contrast, T4, with the highest BAP concentration (1.5 mg L^−1^), significantly reduced shoot number (4.86 c) and internode count (4.58 c), indicating a threshold above which cytokinin becomes inhibitory rather than stimulatory.

These results align with [37], who emphasized the importance of maintaining a balanced hormonal status in in vitro potato cultures. Excessive cytokinins may suppress apical dominance or trigger stress responses that reduce morphogenic efficiency. Our findings support previous observations by [29,30], where moderate hormonal inputs enhanced meristematic activity and cell division. Moreover, the improved root and shoot elongation in T2 and T3 reflect synergistic auxin cytokinin interactions known to govern organogenesis.

Root length also followed this trend, with T2 to T4 outperforming the control, though T4 achieved the longest roots (5.31 cm a). These results agree with [43], who demonstrated that optimized hormonal and environmental parameters in temporary immersion systems (TISs) significantly improved root and shoot morphology in potato explants. Like their findings, our results affirm the effectiveness of specific hormone regimes in enhancing not only shoot multiplication but also underground organ development essential for acclimatization and tuberization. Additionally, internodal development peaked in T1 (control), suggesting that high shoot proliferation in other treatments may reduce spacing between nodes due to compact growth, a tradeoff observed in other micropropagation studies [43,44].

### 3.3. Direct In Vitro Organogenesis in Potato cv. Fianna Improved by Biotic Elicitor Applications

The application of elicitors Activane^®^, Micobiol^®^, and Stemicol^®^ at various concentrations significantly enhanced the morphogenic response of *S. tuberosum* cv. Fianna under in vitro conditions. The percentage of explants responding increased notably across all treatments, ranging from 83% to 98%, compared to only 75% in control. Notably, the shortest time to shoot and root initiation (13 days) was observed in the Stemicol^®^ 0.5 g L^−1^ treatment, indicating its effectiveness in promoting early morphogenesis. These results are visualized in Figure 3 and Figure 4, illustrating treatment-specific differences in callus mass and morphology.

Shoot proliferation was significantly improved across all elicitor treatments compared to the control. The highest shoot numbers were observed in explants treated with Stemicol^®^ at 0.5 g L^−1^ (6.95 a), Micobiol^®^ at 0.5 mL L^−1^ (6.79 a), and Activane^®^ at 0.5 g L^−1^ (6.61 a), all showing a significant advantage over the control (3.84 c). Shoot elongation followed a similar trend, with the longest shoots recorded in Activane^®^ 0.5 (4.57 cm a) and Stemicol^®^ 0.5 (4.28 cm a), suggesting that moderate concentrations of these elicitors enhance shoot architecture. These findings agree with [29,30,35], who demonstrated that intermediate levels of elicitors stimulate endogenous cytokinin-like responses, resulting in enhanced cell division and meristem activity. Likewise, ref. [29] reported improved shoot elongation with elicitor applications, attributing it to increased nutrient uptake and hormonal regulation. Ref. [45] also highlighted the positive impact of certain antimicrobial compounds such as β-lactam antibiotics in promoting shoot organogenesis in potato cultivars, reinforcing the potential dual role of elicitor-type treatments in both microbial suppression and morphogenic stimulation. The limited shoot growth in the control treatment corroborates the observations of [46], who noted restricted shoot proliferation in the absence of external growth regulators.

Root development was also significantly influenced by elicitor treatments. The highest root number (2.83 a) and root length (2.49 cm a) were achieved with Stemicol^®^ at 0.5 g L^−1^, followed by Micobiol^®^ at 0.3 mL L^−1^. In contrast, higher concentrations (0.7 levels) across all elicitors showed diminished root formation, suggesting a threshold beyond which elicitor efficacy declines or becomes inhibitory. These observations align with studies by [37,43], who highlighted the importance of maintaining optimal hormonal balance to promote coordinated shoot and root development. Furthermore, ref. [30] emphasized the role of elicitor-induced hormonal crosstalk in enhancing root organogenesis by modulating auxin signaling pathways. The results indicate that mid-level doses of Micobiol^®^ and Stemicol^®^ are most effective for promoting balanced morphogenesis in *S. tuberosum* cv. Fianna under in vitro conditions, supporting their application in commercial micropropagation systems.

### 3.4. Enhanced Indirect Organogenesis in Potato cv. Fianna Through Elicitor Treatments

The use of elicitors significantly influenced indirect organogenesis in *S. tuberosum* L. cv. Fianna. Notably, Stemicol^®^ and Micobiol^®^ treatments produced superior callogenic responses, as reflected in parameters such as callus initiation time, fresh weight, diameter, and induction percentage. For example, Stemicol^®^ at 0.7 g L^−1^ generated the highest callus fresh weight (5.35 ± 0.17 g) and diameter (6.27 ± 0.18 cm), with 100 percent callus induction. These results are also visualized in Figure 5 and Figure 6, illustrating treatment-specific differences in callus mass and morphology.

These findings align with [38], who emphasized the genotype-dependent response of potato varieties to in vitro culture conditions and highlighted the importance of optimizing culture media to enhance morphogenic potential. Similarly, ref. [47] reported efficient callus induction and shoot regeneration in potato explants under long-day conditions with suitable hormonal treatments, supporting our results that the combination of elicitors and optimized media can accelerate organogenesis. Organogenesis is fundamentally regulated by endogenous hormonal gradients, which are, in turn, influenced by nutrient balance in the culture medium. However, in the current study, no systematic medium optimization was reported; instead, the medium employed contained an unconventional N:P:K ratio and unusual nutrient composition [36]. The sensitivity of indirect organogenesis to hormonal and abiotic factors has been previously documented. Ref. [48] demonstrated that plant growth regulators and salt stress significantly affect callus induction and shoot regeneration in potato, which is consistent with our observation that higher doses of Activane^®^ may induce stress responses, as indicated by delayed callus color change and looser callus texture. Regarding callus quality, Microbiol^®^ treatments at 0.5 and 0.7 mL L^−1^ yielded friable calli, a trait favorable for subsequent shoot regeneration. Ref. [49] reported a similar correlation between friable callus morphology and high-frequency shoot regeneration in *Sansevieria trifasciata*, highlighting the critical role of callus structure in regeneration efficiency.

Furthermore, elicitors have also been shown to enhance secondary metabolite production and biomass accumulation in plants, as reviewed by [39,50] in *Stevia rebaudiana*. Our findings indicate that elicitor treatments not only stimulate morphogenesis but also enhance metabolic activity, which may account for the improved callus vigor observed in our study. The relatively rapid callus initiation within 6 to 8 days demonstrates efficient callogenesis achieved through precise hormone balancing and aligns with previous observations by [47], who reported successful propagation under optimized culture conditions.

## 4. Conclusions

This study demonstrates the effectiveness of biotic elicitors Activane^®^, Micobiol^®^, and Stemicol^®^ in enhancing in vitro morphogenesis of *S. tuberosum* L. cv. Fianna across different stages of micropropagation. The sterilization and culture protocols adopted ensured high asepsis and viability, enabling reproducible shoot and root formation during the establishment phase. Optimal hormonal combinations, particularly 0.2 mg L^−1^ NAA with 0.5–1.0 mg L^−1^ BAP, significantly boosted shoot and root proliferation during the multiplication stage. Notably, mid-range concentrations of Micobiol^®^ and Stemicol^®^ (0.5 mL/g L^−1^) were most effective in promoting direct organogenesis, reducing initiation time and enhancing shoot-root architecture. Indirect organogenesis was also improved, with Stemicol^®^ at 0.7 g L^−1^ producing robust, compact calli with the highest fresh weight and diameter. These findings highlight the dual role of elicitors in stimulating morphogenic responses and potentially enhancing metabolite production, offering a promising approach to replacing or supplementing conventional growth regulators. The integration of elicitor treatments into micropropagation protocols can increase efficiency, reduce costs, and contribute to sustainable in vitro propagation systems for commercial potato production and germplasm conservation.

## Figures and Tables

**Figure 1 biotech-14-00077-f001:**
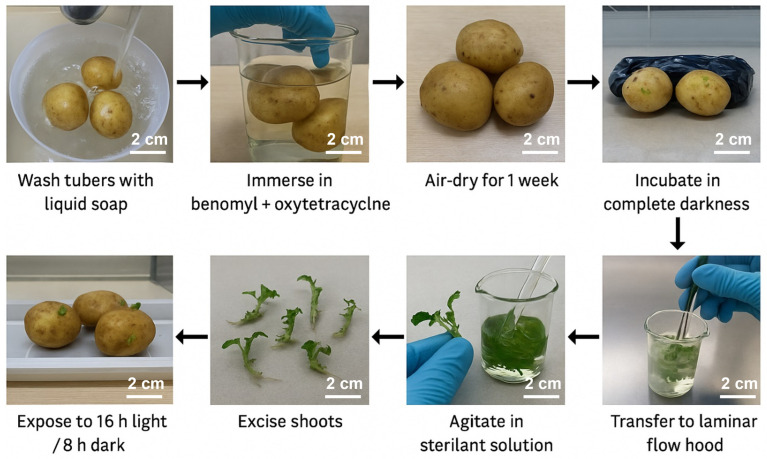
Flowchart illustrates the detailed protocol for the preparation of potato (*S. tuberosum* L. cv. Fianna) tubers and excised shoots for in vitro multiplication and organogenesis.

**Figure 2 biotech-14-00077-f002:**
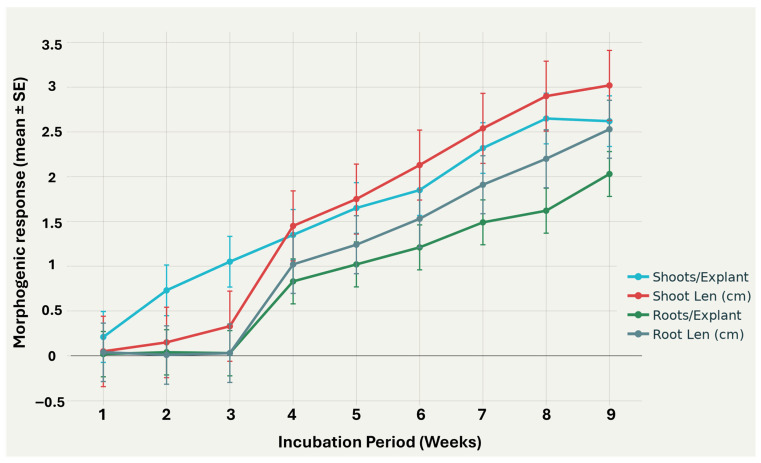
Linear representation of the morphogenic response of *S. tuberosum* cv. “Fianna” under in vitro culture conditions over nine weeks.

**Figure 3 biotech-14-00077-f003:**
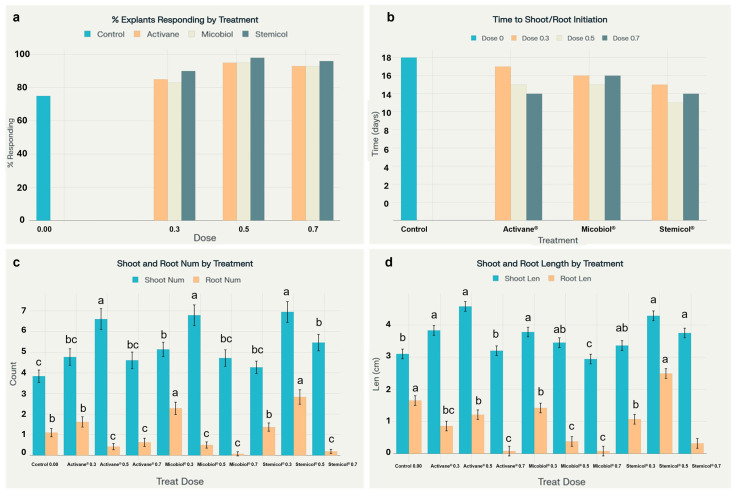
Elicitor treatments enhanced in vitro regeneration of *S. tuberosum* cv. Fianna. (**a**) Explant response was highest with Stemicol^®^ 0.5 g L^−1^ (98%) compared to the control (75%). (**b**) Time to shoot/root initiation was shortest under Stemicol^®^ 0.5 (13 days). (**c**) The greatest shoot and root numbers were observed in Stemicol^®^ 0.5 (6.95 shoots, 2.83 roots) and Micobiol^®^ 0.3 (2.28 roots). (**d**) Shoot and root lengths were maximized with Activane^®^ 0.5 (4.57 cm) and Stemicol^®^ 0.5 (2.49 cm), respectively. Statistical differences are indicated by different letters (Tukey’s HSD, *p* < 0.05).

**Figure 4 biotech-14-00077-f004:**
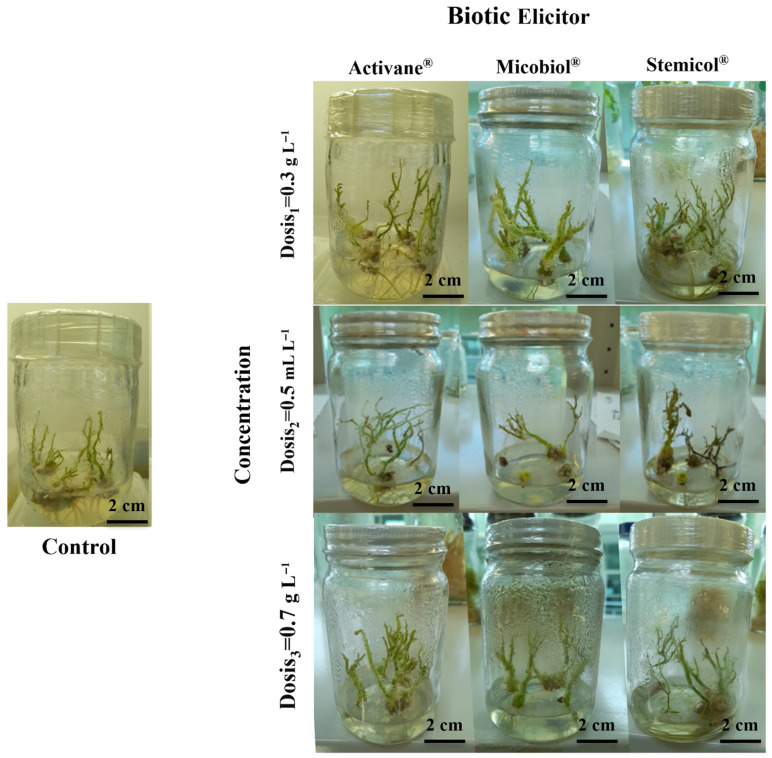
Direct in vitro organogenesis in *S. tuberosum* cv. Fianna improved by biotic elicitor applications. Representative shoot development after four weeks of culture under different elicitor treatments: Control, Activane^®^ 0.5 g L^−1^, Micobiol^®^ 0.5 mL L^−1^, and Stemicol^®^ 0.5 g L^−1^. Enhanced shoot proliferation and elongation are clearly observed in elicitor-treated explants compared to the control, indicating the positive effect of biotic elicitors on direct organogenesis.

**Figure 5 biotech-14-00077-f005:**
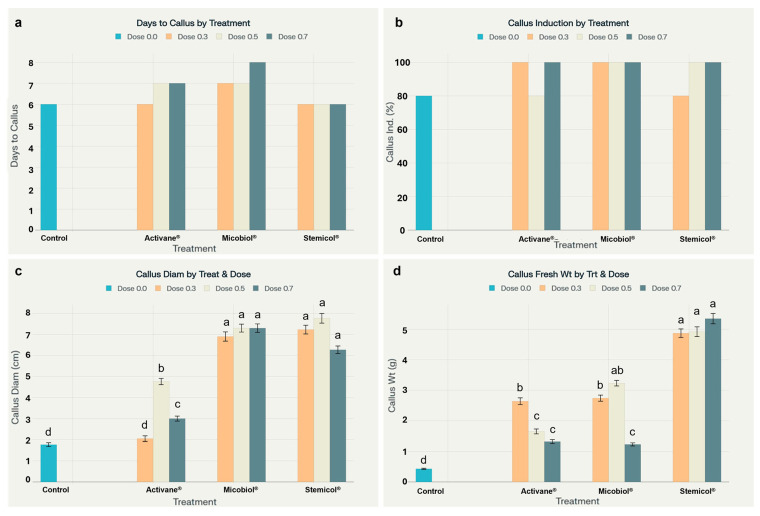
Elicitor treatments improved callus development in *S. tuberosum* cv. Fianna. (**a**) Days to callus formation were consistent across treatments. (**b**) Callus induction reached 100% in most treated groups. (**c**) Callus diameter was significantly larger in Micobiol^®^ and Stemicol^®^ treatments. (**d**) Callus fresh weight peaked under Stemicol^®^, significantly exceeding the control. Different letters indicate significant differences (Tukey’s HSD, *p* < 0.05).

**Figure 6 biotech-14-00077-f006:**
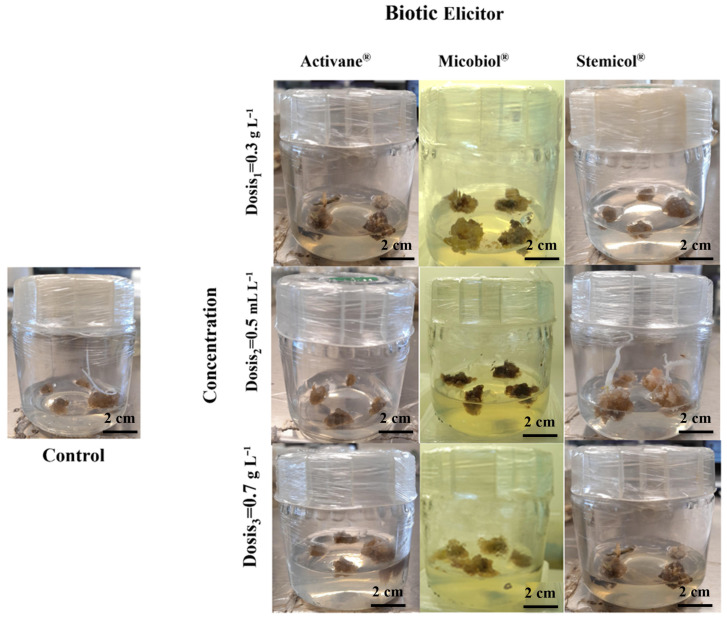
Enhanced indirect organogenesis in *S. tuberosum* cv. Fianna through elicitor treatments. Representative callus formation from explants cultured under different elicitor treatments: Control, Activane^®^ 0.5 g L^−1^, Micobiol^®^ 0.5 mL L^−1^, and Stemicol^®^ 0.5 g L^−1^. Elicitor treatments, particularly Micobiol^®^ and Stemicol^®^, resulted in more vigorous, compact, and friable calli with greater biomass compared to the control. These visual differences reflect the stimulatory effect of elicitors on callogenesis and the potential for improved regeneration efficiency.

**Table 1 biotech-14-00077-t001:** Effect of different concentrations of BAP and NAA on in vitro shoot multiplication of *S. tuberosum* L. cv. Fianna.

Treatment	Hormone	Shoot Number	Shoot Length (cm)	Root Number	Root Length (cm)	Internode
T1:	Control (0.0)	5.00 b	4.45 b	5.10 b	4.20 b	6.58 a
T2:	0.2 NAA + 0.5 BAP	6.95 a	6.98 a	6.10 a	5.21 a	5.48 ab
T3:	0.2 NAA + 1.0 BAP	6.16 a	6.90 a	6.10 a	5.15 a	5.16 ab
T4:	0.2 NAA + 1.5 BAP	4.86 c	4.73 b	5.20 b	5.31 a	4.58 c

Note: Different letters within each column indicate statistically significant differences according to Tukey’s test (*p* ≤ 0.05).

## Data Availability

The original contributions presented in this study are included in the article. Further inquiries can be directed to the corresponding author(s).

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
