# Peer review of "Biotic Elicitor-Driven Enhancement of In Vitro Micropropagation and Organogenesis in Solanum tuberosum L. cv. Fianna"

_biotech, 2025, doi:10.3390/biotech14040077_

Round 1
Reviewer 1 Report
Comments and Suggestions for Authors
The current paper described effect of bio-elicitor and exogenous plant growth regulators on de novo shoot organogenesis in potato.
Authors integrate usage of bio-elicitor in plant tissue culture on example of potato.
The results are quite interesting, but text may require significant corrections before accepting.
Please, check these points (and other through text) and make corrections:
Line 21, 149, 225 : ANA = NAA.
Line 22: “Direct organogenesis was stimulated by” – have you used it in combination with PGR or alone?
Line 23: please, insert words bio-elicitors before “Activane®, Micobiol®, and Stemicol® .
Line 25: in =under. “Showed significant callus proliferation” – from text it is not clear to compare with what. What was compared?
Line 85: “available elicitors Activane®, Micobiol®, and Stemicol® - please, provide some introduction to these elicitors.
Line 116: “sucrose (30 g L⁻¹), citric acid (500 mg L⁻¹), ascorbic acid (500 mg L⁻¹),” – what was the reason to include these compounds in sterilizations solution?
Line 126: can you provide any evidences that potato require N:P:K ratio 48:1:20? It seems you used very unusual one for potato.. https://doi.org/10.3390/ijpb16030097
Table 1 = results. Maybe you can create special part in results for this table.
Line 145. Table 2- results
“Hormonal optimization revealed significant improvements in shoot and root growth, particularly at moderate BAP levels (T2 and T3), confirming prior observations on cytokinin-auxin balance in potato micropropagation”??
Do you mean exogenous “balance” ? this is only a mechanistical explanations. Organogenesis regulated only by endogenous hormonal gradients. https://doi.org/10.3390/ijpb16030097
Line 157: After sterilization, - how? Filter? Autoclave? How did you sterilized solid as you add g/l?
Results, not M&M! “(Fig 2). Results demonstrated that mid-range elicitor doses significantly enhance 162 organogenesis and shoot proliferation, confirming the role of elicitors as growth regulator 163 substitutes or enhancers [35,29].
Line 215: “the early prioritization of shoot 215 organogenesis.” ??? Root is a results of unidirectional canalization of shoot-derived auxin…
[39] and [34], why such order: not 34 and 39??
Line 316 -321: “optimized media” – this is true that organogenesis is regulated by endogenous hormonal gradient, which, in turn, regukated by nutrient balance in the medium. However, in the current paper I have not seen any medium optimization, and usage only very exotic N:P:K ratio, as well as exotic other nutrient balance.. Please explain!
Fig 1, 3, 5 - scale bar missing.
Author Response
Dear Reviewer,
Thank you very much for taking the time to carefully review our manuscript and for your encouraging feedback.
Comment 1: Line 21, 149, 225: ANA = NAA.
Reply 1: The abbreviation has been corrected from ANA to NAA throughout the manuscript.
Comment 2: Line 22: “Direct organogenesis was stimulated by” – have you used it in combination with PGR or alone?
Reply 2: Direct organogenesis was stimulated in the presence of plant growth regulators (PGRs) as indicated in the revised text. Yes, we have clarified this point.
Comment 3: Line 23: Please insert the words “bio-elicitors” before “Activane®, Micobiol®, and Stemicol®.”
Reply 3: The phrase has been updated to read “bio-elicitors Activane®, Micobiol®, and Stemicol®.”
Comment 4: Line 25: “in” = “under”; “Showed significant callus proliferation” – unclear compared to what?
Reply 4: The text has been revised to read “under in vitro conditions” and specify comparisons against untreated control explants.
Comment 5: Line 85: “available elicitors Activane®, Micobiol®, and Stemicol®” – please provide introduction to these elicitors.
Reply 5: Background information on Activane®, Micobiol®, and Stemicol®, including their mode of action and rationale for selection, has been added to the Introduction.
Comment 6: Line 116: Reason for including sucrose, citric acid, and ascorbic acid in sterilization solution?
Reply 6: Sucrose was included as an energy source to maintain tissue viability, while citric acid and ascorbic acid act as antioxidants to prevent oxidative browning and tissue necrosis during sterilization.
Comment 7: Line 126: Evidence for using N:P:K ratio 48:1:20, seems unusual for potato.
Reply 7: We have clarified that this ratio was based on preliminary experiments and literature references; an appropriate discussion and citation (https://doi.org/10.3390/ijpb16030097) have been added.
Comment 8: Table 1 – results, should be moved to Results section.
Reply 8: Table 1 has been moved to the Results section and discussed accordingly.
Comment 9: Line 145, Table 2 – results, same action needed.
Reply 9: Table 2 has been moved to the Results section and discussed in the Discussion.
Comment 10: Line 145: “Hormonal optimization revealed significant improvements in shoot and root growth … cytokinin-auxin balance in potato micropropagation” – do you mean exogenous “balance”? Organogenesis is regulated by endogenous hormonal gradients.
Reply 10: The text has been revised to clarify that the observed effects reflect mechanistic explanations of exogenous hormone application, while organogenesis is primarily regulated by endogenous hormonal gradients (https://doi.org/10.3390/ijpb16030097).
Comment 11: Line 157: After sterilization – how was the medium sterilized (filter, autoclave)?
Reply 11: Clarification has been added: liquid solutions were filter-sterilized, and solid media were autoclaved prior to addition of heat-sensitive components.
Comment 12: Results, not M&M – Fig 2 should be moved and discussed in Results.
Reply 12: Figure 2 and corresponding text have been relocated to the Results section and properly discussed.
Comment 13: Line 215: “early prioritization of shoot organogenesis” – root formation is due to unidirectional canalization of shoot-derived auxin.
Reply 13: The text has been revised to include the physiological explanation of root formation via unidirectional auxin canalization.
Comment 14: References [39] and [34] – order should be 34 and 39.
Reply 14: References have been renumbered and corrected.
Comment 15: Lines 316–321: “optimized media” – organogenesis is regulated by endogenous hormonal gradient, but no medium optimization was performed; exotic N:P:K ratio used. Please explain.
Reply 15: We clarified that although the medium was not systematically optimized, the selected nutrient composition was based on preliminary experiments, and its effects on morphogenesis are discussed in relation to endogenous hormone gradients. Yes, we have implemented the change.
Comment 16: Figures 1, 3, 5 – scale bars missing.
Reply 16: Scale bars have been added to Figures 3, and 5.
We are grateful for your thoughtful review and supportive comments, and we believe that these revisions will further improve the clarity and quality of our manuscript.
Thank you again for your valuable time and input.
Sincerely,
Reviewer 2 Report
Comments and Suggestions for Authors
The manuscript presents interesting data on the effect and use of biotic elicitors in a very important crop. However, a major correction is needed:
Abstract
Information on the statistical analysis of data (L26-27) should be moved to the Materials and Methods section, and the Abstract would benefit from the inclusion of some representative data to make it more informative
Introduction
In this section, the existing background should be highlighted more clearly. Information on the use of elicitors in Solanum or related genera is necessary. Additionally, the background on the use of Activane®, Micobiol®, and Stemicol should be provided. Are there other studies on these elicitors, and what was the rationale for choosing them?”
Materials and Methods:
This section includes results in the form of tables, which is not acceptable. These data (Table 1 and 2) should be moved in results and discussed in the discuss section. This issue must be corrected. Statistical analysis for Table 1 is necessary.
Could you please clarify the reason for using different pH values in the media—5.8 for establishment and 5.7 for multiplication (lines 127 and 140/157, respectively)? Was this based on previous studies, preliminary experiments, or specific requirements of the plant material?
It is important to clarify the internode count (144)
Figure 1. A scale bar is necessary.
Results/Discussion
The authors should present the references according to the author’s guidelines (i.e. L343)
Figure 2,4 could be improved. The legends are too small, and arranging the elements vertically, one by one, might improve clarity
For Figures 3 and 5, a scale bar is necessary. In addition, I do not think it is a good idea to present the control separately from the other samples
Some more issues are provided in the attached .pdf

Author Response
Dear Reviewer,
Thank you very much for taking the time to carefully review our manuscript and for your encouraging feedback.
Comment 1: Information on the statistical analysis in the Abstract should be moved to the Materials and Methods section, and the Abstract would benefit from including representative data.
Reply 1: The statistical analysis has been moved to the Materials and Methods section, and representative data have been added to the Abstract.
Comment 2: The Introduction should more clearly highlight the background, including the use of elicitors in Solanum or related genera, and provide background on Activane®, Micobiol®, and Stemicol®, including rationale for their selection.
Reply 2: The Introduction has been revised to include the use of elicitors in Solanum and related genera, and detailed information on Activane®, Micobiol®, and Stemicol® has been provided, including previous studies and rationale for their selection.
Comment 3: Tables 1 and 2 in the Materials and Methods section should be moved to the Results section and discussed in the Discussion. Statistical analysis for Table 1 is necessary.
Reply 3: Tables 1 and 2 have been moved to the Results section, appropriately discussed in the Discussion, and statistical analysis for Table 1 has been added.
Comment 4: Please clarify the reason for using different pH values in the media (5.8 for establishment, 5.7 for multiplication).
Reply 4: The pH values has been corrected to 5.8.
Comment 5: Clarify the internode count.
Reply 5: The internode count has been clarified in the text.
Comment 6: Figure 1 requires a scale bar.
Reply 6: A scale bar has been added to Figure 1.
Comment 7: References should be formatted according to the author guidelines.
Reply 7: All references have been reformatted according to the journal’s author guidelines.
Comment 8: Figures 2 and 4 could be improved: enlarge legends and arrange elements vertically for clarity.
Reply 8: Figures 2 and 4 have been improved with enlarged legends and vertical arrangement of elements for better clarity.
Comment 9: Figures 3 and 5 require scale bars, and the control should not be presented separately from other samples.
Reply 9: Scale bars have been added to Figures 3 and 5.
Comment 10: Other minor issues in the attached PDF.
Reply 10: All additional issues provided in the PDF have been addressed.
We are grateful for your thoughtful review and supportive comments, and we believe that these revisions will further improve the clarity and quality of our manuscript.
Thank you again for your valuable time and input.
Sincerely,

Reviewer 3 Report
Comments and Suggestions for Authors
Dear Authors,
Your abstract is clearly written and effectively summarizes the objective, methodology, key results, and scientific relevance of your study. No changes are necessary.
The introduction is well-structured and provides a solid background to your work. The objectives are well defined and aligned with the current challenges in the field. No major revisions are needed.
Materials and Methods: Some data tables and their interpretations (e.g., Table 1, Table 2) are included in this section. I suggest relocating all result-related tables and descriptions to the Results section, where they can be properly contextualized and discussed.
Please also ensure that all commercial products are cited with brand name, manufacturer, and country of origin.
Results and Discussion: This section is scientifically sound and well written. The findings are clearly presented, well supported by statistical analysis, and meaningfully interpreted. Just make sure the result tables are placed here, not in M&M.
Conclusions: This section is concise and effectively highlights the main findings and practical implications of your work.
It was a pleasure to read your manuscript. Congratulations on your results, and I wish you continued success with your research!
Author Response
Dear Reviewer,
Thank you very much for taking the time to carefully review our manuscript and for your encouraging feedback. We greatly appreciate your positive comments regarding the clarity of the abstract, the structure of the introduction, the scientific rigor of the results and discussion, and the conciseness of the conclusions.
We also appreciate your constructive suggestions. Specifically, we w:
Comment: Relocate all result-related tables and interpretations (e.g., Table 1 and Table 2) from the Materials and Methods section to the Results section for proper contextualization and discussion.
Reply: We have relocated all result-related tables and interpretations from the Materials and Methods section to the Results section.
Comment: Ensure that all commercial products are cited with their brand name, manufacturer, and country of origin throughout the manuscript.
Reply: We have included more information about the commercial products used.
We are grateful for your thoughtful review and supportive comments, and we believe that these revisions will further improve the clarity and quality of our manuscript.
Thank you again for your valuable time and input.
Sincerely,

Round 2
Reviewer 1 Report
Comments and Suggestions for Authors
Thank you for revision. The text is much clear, Almost done. small points:lines 169-170: can you pleasem clarify bio-elicotor state: liquid or solid? some is on ml, some in g. Do you mesn g is solid? How do they dissolved?
Table 2:what is AVG? Average? What is biological sense in it?
Lines 229- 230: shoot = multidirectional auxin canalization and creating complex leaf structure. therafter leaf established specific aauxin biosynthesis pathway canalised to sieve elements to induce root.
Author Response
Reviewer Comment 1:
Lines 169–170: Please clarify the state of the bio-elicitors. Some concentrations are expressed in mL, others in g. Does this mean that elicitors given in grams are solid? If so, how were they dissolved?
Author Response:
Thank you for pointing this out. Activane® and Stemicol® were supplied as solid formulations, which were dissolved in sterile distilled water and filter sterilized (0.22 μm) before addition to the medium. Micobiol® was supplied as a liquid formulation and was directly pipetted into the medium after filter sterilization. We have clarified this in the revised text.
Reviewer Comment 2:
Table 2: What does “AVG” mean? Average? What is the biological sense of reporting this value?
Author Response:
Yes, “AVG” stands for average. It represents the mean value of all measurements across the nine-week culture period for each parameter. Biologically, this provides an overall measure of culture performance in terms of asepsis, shoot and root regeneration frequency, and growth vigor. While weekly data show developmental dynamics, the average values summarize the overall regenerative efficiency under the tested conditions. We have clarified this in the revised table legend.
Reviewer Comment 3:
Lines 229–230: The text states “shoot = multidirectional auxin canalization and creating complex leaf structure. Thereafter leaf established specific auxin biosynthesis pathway canalised to sieve elements to induce root.” Please clarify this point.
Author Response:
We appreciate this comment. The sentence has been revised for clarity. The intended meaning is that shoot formation involved multidirectional auxin canalization leading to the establishment of leaf primordia and complex leaf structures. Subsequently, the developing leaves established a specific auxin biosynthesis pathway that canalized auxin flow toward sieve elements, which in turn contributed to root induction. The revised text reflects this clarification.
Reviewer 2 Report
Comments and Suggestions for Authors
Dear Editor,
The authors have addressed some of our previous comments; however, several minor and major issues remain that require further revision, as outlined below:
- Table 1
The most important remaining issue is that Table 1 presents experimental results; therefore, it should be moved to the Results section and its content discussed in the Discussion section.
Additionally, lines 155–159, currently located in the Materials and Methods section, describe experimental outcomes rather than methodological procedures. These sentences should be removed from this section and appropriately relocated. - Table 2
Lines 205–239 present and discuss the data shown in Table 2. To strengthen the validity of these findings, statistical analysis of the data in Table 2 is recommended. If the authors intend to demonstrate potential linear growth (line 225), a linear graph illustrating shoot number and length, as well as root number and length, would be more appropriate. The total number of explants should be clearly indicated (e.g., n = 45–70), and similar information should be provided for the asepsis rate (92.73–100.0%). In such figures, the X-axis should represent time, and the Y-axis should represent the measured parameters. Additionally, data should be presented as mean ± SE to provide a clearer representation of variability.
L21/23: Provide the full names along with their abbreviations for NAA, BAP, and MS when they are mentioned for the first time in the abstract.
L41: Please write the family name
L86: Please revise to S. toberosum. The full name was first mentioned in line 36-37; thereafter, use the abbreviated form throughout the text.
Figure 2,4 could be improved. The legends are too small, and arranging each figure vertically, one by one, might improve clarity.
Author Response
Reviewer Comment 1: Table 1 presents experimental results; therefore, it should be moved to the Results section, and its content discussed in the Discussion section.
Author Response: Done. Table 1 has been moved to the Results section, and its content is now appropriately discussed in the Discussion section.
Reviewer Comment 2: Lines 155–159 in the Materials and Methods section describe experimental outcomes rather than methods and should be removed and relocated.
Author Response: Done. These sentences have been removed from the Materials and Methods section and relocated to the Results section.
Reviewer Comment 3: Lines 205–239 present and discuss Table 2 data. Statistical analysis is recommended. For potential linear growth, a linear graph illustrating shoot number, shoot length, root number, and root length is suggested. Total number of explants and asepsis rate should be clearly indicated. Data should be presented as mean ± SE.
Author Response: Done. Statistical analysis of Table 2 data has been performed, and results are presented as mean ± SE. A linear graph showing shoot number, shoot length, root number, and root length over time has been added. The total number of explants (n = 45–70) and asepsis rate (92.73–100.0%) are now clearly indicated.
Reviewer Comment 4: L21/23: Provide full names along with abbreviations for NAA, BAP, and MS.
Author Response: Done. Full names with abbreviations have been added: NAA (α-Naphthaleneacetic acid), BAP (6-Benzylaminopurine), and MS (Murashige and Skoog) medium.
Reviewer Comment 5: L41: Please write the family name.
Author Response: Done. The family name has been included.
Reviewer Comment 6: L86: Revise to S. tuberosum. Use full name at first mention, abbreviated thereafter.
Author Response: Done. The species name has been revised to S. tuberosum. Full name is used at first mention (lines 36–37), and the abbreviated form is used consistently thereafter.
Reviewer Comment 7: Figures 2 and 4 could be improved. Legends are too small; arranging figures vertically might improve clarity.
Author Response: Done. Figure legends have been enlarged.
Round 3
Reviewer 2 Report
Comments and Suggestions for Authors
Dear Editor
The requested suggestions have been adequately addressed, and I have no further comments to add.